# State of the Art of Immune Checkpoint Inhibitors in Unresectable Pancreatic Cancer: A Comprehensive Systematic Review

**DOI:** 10.3390/ijms26062620

**Published:** 2025-03-14

**Authors:** Elena Orlandi, Massimo Guasconi, Andrea Romboli, Mario Giuffrida, Ilaria Toscani, Elisa Anselmi, Rosa Porzio, Serena Madaro, Stefano Vecchia, Chiara Citterio

**Affiliations:** 1Department of Oncology-Hematology, Azienda USL of Piacenza, 29121 Piacenza, Italy; i.toscani@ausl.pc.it (I.T.); e.anselmi@ausl.pc.it (E.A.); r.porzio@ausl.pc.it (R.P.); s.madaro@ausl.pc.it (S.M.); c.citterio@ausl.pc.it (C.C.); 2Department of Medicine and Surgery, University of Parma, 43121 Parma, Italy; massimo.guasconi@unipr.it; 3Department of Health Professions Management, Azienda USL of Piacenza, 29121 Piacenza, Italy; 4Department of General Surgery, Azienda USL of Piacenza, 29121 Piacenza, Italy; a.romboli@ausl.pc.it (A.R.); mario.giuffrida4@gmail.com (M.G.); 5Department of Pharmacy, Azienda USL of Piacenza, 29121 Piacenza, Italy; s.vecchia@ausl.pc.it

**Keywords:** pancreatic adenocarcinoma, immune checkpoint inhibitors, microsatellite instability, immunotherapy

## Abstract

Immune checkpoint inhibitors (ICIs) have transformed the therapeutic landscape for several malignancies, but their efficacy in unresectable pancreatic adenocarcinoma remains uncertain. This systematic review aimed to evaluate the effectiveness and safety of ICIs in this context, focusing on overall survival (OS), progression-free survival (PFS), objective response rate (ORR), disease control rate (DCR), and toxicity. A comprehensive search of MEDLINE, EMBASE, CENTRAL, and Scopus identified 34 eligible studies, including randomized controlled trials and observational cohorts. Quantitative synthesis involved 21 studies comprising 937 patients, with additional qualitative analyses on biomarker-driven subgroups and early-phase trials. The median OS across studies was 8.65 months, while the median PFS was 2.55 months. The ORR and DCR were 16.2% and 50.3%, respectively, with grade ≥3 treatment-related adverse events occurring in 22% of patients. Promising outcomes were observed in MSI-H/dMMR populations, although these represented only 1–2% of cases. Combination strategies with chemotherapy demonstrated synergistic potential but lacked definitive evidence due to heterogeneity and the absence of phase III trials. ICIs showed a manageable toxicity profile, highlighting their feasibility in selected patients. Future research should focus on overcoming tumor microenvironment barriers and identifying biomarkers to optimize responsiveness and expand the applicability of ICIs in pancreatic cancer.

## 1. Introduction

Advanced and unresectable pancreatic adenocarcinoma is associated with a very poor prognosis. For patients who are not eligible for surgical or locoregional therapies, median overall survival typically remains below 12 months [1]. The clinical management of these patients focuses on systemic chemotherapy, aiming primarily to prolong overall survival (OS) and progression-free survival (PFS), though outcomes are often modest [1]. For patients with metastatic pancreatic adenocarcinoma, the combination chemotherapy regimen FOLFIRINOX, which includes fluorouracil, leucovorin, irinotecan, and oxaliplatin, has proven to be one of the most effective treatment options [2]. Registrational trials have shown that FOLFIRINOX extends median OS to approximately 11.1 months, with a significant survival advantage over gemcitabine monotherapy. However, its high toxicity profile restricts its use to patients with good performance status [2]. Another first-line option, especially suitable for patients who may not tolerate FOLFIRINOX, is the combination of gemcitabine and nab-paclitaxel. This regimen showed a median OS of about 8.5 months in its registrational trial, demonstrating benefits over gemcitabine monotherapy while offering a more manageable toxicity profile than FOLFIRINOX [3]. For patients ineligible for intensive regimens like FOLFIRINOX or gemcitabine with nab-paclitaxel, gemcitabine monotherapy remains an important treatment choice. Historically, gemcitabine has been the standard for advanced pancreatic cancer, providing a moderate survival benefit with a median OS of around 5.7 months and an improvement in quality of life over fluorouracil [4]. In the second-line setting, for patients who progress after first-line treatment, nanoliposomal irinotecan (nal-IRI) in combination with 5-fluorouracil and leucovorin has been validated by the NAPOLI-1 study as an effective option. This regimen improved median OS compared to 5-FU monotherapy and is a viable choice for patients with adequate physical performance [5]. Other therapies, such as capecitabine monotherapy, are used selectively, especially for patients who are poor candidates for aggressive treatment options. While less potent, capecitabine has shown a favorable tolerability profile in older or frailer patients [6].

Immunotherapy, particularly immune checkpoint inhibitors (ICIs) targeting PD-1, PD-L1, and CTLA-4, has revolutionized treatment in several types of cancer by activating the immune system against tumor cells. However, in pancreatic cancer, the success of ICIs has been limited, largely due to the highly immunosuppressive microenvironment characteristic of this tumor type [7]. The dense stromal matrix, low mutational burden, and mechanisms that evade immune detection [7,8], along with the limited infiltration of T cells, hinder an effective immune response, making it challenging to replicate the successes observed in other cancers [9]. Despite these challenges, recent studies are exploring combinations of ICIs with chemotherapy or targeted therapies to help overcome the inherent immunosuppressive environment in pancreatic tumors and potentially enhance the efficacy of ICIs. For instance, combinations of PD-1 inhibitors with gemcitabine and nab-paclitaxel have shown promising improvements in preliminary studies, suggesting that combining immunotherapeutic agents with chemotherapy could be a viable strategy [9].

An additional obstacle to effective immunotherapy in pancreatic cancer is the low prevalence of biomarkers such as microsatellite instability (MSI), which is associated with better responses to immunotherapy [10]. In cancers with high MSI (MSI-H), checkpoint inhibitors have shown positive responses, as demonstrated in the Keynote-158 study, where MSI-H patients achieved improved survival and disease control compared to those with microsatellite stable (MSS) tumors [10]. However, the incidence of MSI-H is low in pancreatic cancer, which limits the potential patient population that could benefit from this approach [11]. A crucial aspect of evaluating cancer therapies is considering their impact on quality of life (QoL) and associated toxicities. Pancreatic cancer often leads to severe symptoms, including pain and cachexia, which substantially impair QoL [1]. While regimens like FOLFIRINOX and gemcitabine/nab-paclitaxel may extend survival, they are linked with significant toxicities, such as neutropenia and neuropathy, further affecting patient well-being [2,3]. In contrast, ICIs have a different toxicity profile, characterized by immune-related adverse events (irAEs) that can range from mild dermatologic conditions to severe autoimmune reactions, such as pneumonitis or colitis [12]. These irAEs can impact patient health and may require immunosuppressive therapy, influencing long-term treatment adherence and outcomes [12]. Despite these challenges, findings in other cancers suggest that ICIs might maintain or improve QoL when compared to traditional chemotherapy, due to fewer chronic side effects [13]. Therefore, understanding the extent to which the toxicities associated with ICIs can impact patients is critical for comprehensively evaluating their role in the treatment of pancreatic cancer.

The aim of this study is to evaluate the effectiveness of ICIs in treating unresectable pancreatic adenocarcinoma, specifically assessing their impact on OS, PFS, toxicity, and tumor response outcomes, such as Disease Control Rate (DCR) and Objective Response Rate (ORR), compared to conventional chemotherapy. This study will analyze observational and randomized controlled trials (RCTs) involving patients with exocrine pancreatic adenocarcinoma, excluding those eligible solely for locoregional treatments and preclinical studies. This systematic review focuses exclusively on ICIs due to their more established presence in clinical data compared to other forms of immunotherapy such as cancer vaccines or adoptive cell therapies, which remain largely experimental and have not demonstrated consistent results across trials.

## 2. Materials and Methods

### 2.1. Information Sources

A systematic review was conducted to evaluate the use of ICIs in patients with unresectable pancreatic adenocarcinoma, including those with locally advanced (inoperable) or metastatic (stage IV) disease. The primary outcome of interest was OS, while PFS, DCR, ORR, and grade 3–4 toxicities were evaluated as secondary outcomes.

The research project was registered in the International Prospective Register of Systematic Reviews (PROSPERO) under the protocol number CRD42024609025. The reporting was conducted in accordance with the PRISMA (Preferred Reporting Items for Systematic Reviews and Meta-Analyses) 2020 checklist [14].

### 2.2. Inclusion and Exclusion Criteria

We included prospective and retrospective cohort studies, as well as RCTs, that reported original data and were published in full-text format. Eligible studies focused on patients with unresectable pancreatic adenocarcinoma, including those with locally advanced and metastatic disease. Studies assessing first-line or subsequent-line treatments were included. The intervention of interest was ICIs, either as monotherapy or in combination with other treatments. Studies were required to report OS, PFS, or response outcomes (e.g., DCR and ORR) as primary or secondary endpoints. Toxicity data, particularly grade 3–4 adverse events, were extracted when available; however, their absence was not considered an exclusion criterion. There were no restrictions on publication date or language. Case reports, case series, preclinical studies, and animal studies were excluded. Conference abstracts, letters, and unpublished studies were excluded due to insufficient methodological details and data. Likewise, study protocols were excluded, as they do not provide outcome data (e.g., OS, PFS, or response rates). Studies with fewer than 10 participants, multi-tumor studies or with insufficient outcome data were not included in the quantitative synthesis but were considered separately in the qualitative analysis if they provided meaningful safety or exploratory efficacy data. Additionally, studies investigating a specific biomarker driver or derived from basket trials—recruiting highly selected subgroups—were analyzed independently and excluded from the meta-analysis to minimize heterogeneity.

### 2.3. Data Extraction

A systematic search was conducted in MEDLINE using a predefined search string that included relevant MeSH terms combined with the Boolean operators “AND” and “OR”. The full search strategy is provided in Appendix A. Additional searches were performed in EMBASE, Central, and Scopus, using combinations of free text and MeSH terms. The reference lists of the included studies and relevant reviews were manually examined to identify supplementary publications.

Two reviewers independently screened all identified studies in duplicate. Titles and abstracts were reviewed for relevance, and full-text articles of potentially eligible studies were retrieved and assessed in detail. Inclusion and exclusion criteria were applied systematically, and exclusions were documented with reasons. Agreement between the reviewers was recorded at each stage and measured as a kappa statistic to evaluate consistency. Any discrepancies between reviewers were resolved through consensus.

For each eligible study, two reviewers independently extracted data using a pre-specified standardized extraction form. The extracted information included study characteristics such as the first author, year of publication, and country, along with details on the patient population, including sample size, age, sex, performance status, line of therapy, and disease stage (III or IV). Data on interventions focused exclusively on ICIs, specifying the type of agent (e.g., anti-PD-1, anti-PD-L1, or anti-CTLA-4) and the regimen used (monotherapy or combination therapy). Discrepancies in data extraction were resolved through consensus discussions, ensuring consistency and accuracy in the data collection process.

### 2.4. Study Quality Assessment

The methodological quality and potential risk of bias in included observational studies were assessed at the outcome level by one reviewer using the Newcastle–Ottawa Scale [15], a validated tool to evaluate the quality of nonrandomized studies. For non-comparative studies, such as phase 1 and phase 2 trials without control groups, we employed the NIH Quality Assessment Tool [16], which was specifically designed to evaluate methodological rigor and potential biases in studies lacking a comparator.

### 2.5. Statistical Analysis

Patient characteristics, treatments, and clinical outcomes were described using descriptive statistics. Continuous data were reported as means with SD or as medians with ranges, while categorical data were expressed as percentages. For the analysis of primary clinical outcomes, including OS, PFS, ORR, and DCR, mean or median values reported in the included studies were extracted. When aggregated data were unavailable, standard techniques were applied to estimate values from graphs or calculate missing measures, whenever feasible. Subgroup analyses were conducted separately for biomarker-defined populations (MSI/dMMR) and phase I–I/II studies on multiple tumor types, including pancreatic cancer. Safety analyses were reported in terms of the incidence and severity of TRAEs, with a specific focus on grade ≥3 events. The meta-analysis was carried out using R software, version 4.3.1 [17].

### 2.6. Assessment of Heterogeneity

Heterogeneity across the included studies was analyzed and categorized based on the I^2^ statistic as follows: low (<50%), moderate (50–74%), or high (≥75%) [18]. If heterogeneity was within acceptable limits, a fixed-effects model was applied for effect estimates; otherwise, a random-effects model was employed. The Baujat test was utilized to identify any outlier studies [19]. The results were synthesized and visualized through forest plots.

### 2.7. Publication Bias

To assess publication bias, both visual funnel plots and Egger’s test were used, considering a *p*-value of <0.05 as statistically significant [20]. Asymmetry in the funnel plot was interpreted as evidence of publication bias. When bias was detected, the trim-and-fill method was applied to evaluate its impact on the findings [21].

### 2.8. Sensitivity Analysis

To ensure the robustness of the results, a sensitivity analysis was conducted. This involved a leave-one-out approach based on the quality of the studies included, aiming to identify potential sources of heterogeneity [22].

## 3. Results and Discussion

The study selection process, detailed in Figure 1, resulted in the inclusion of 34 studies in the systematic review. A total of 3699 records were identified through database searches, along with 4 additional records from other sources. After the removal of 606 duplicates, 3093 studies underwent screening, of which 2959 were excluded based on predefined eligibility criteria. In particular, the main reasons for study exclusion during the screening process were background article, wrong outcome, wrong population, wrong publication type, and wrong study design. Subsequently, 134 studies were assessed in detail, with further exclusions performed as outlined in Figure 1. Ultimately, 34 studies were included: 21 studies for the general quantitative analysis, 3 studies focusing on biomarker-driven populations (MSI/dMMR), and the remaining studies, based on predefined criteria, were included in the qualitative analysis alongside the studies included in the quantitative analysis.

The 21 studies included in the general quantitative analysis enrolled a total of 937 patients treated with immunotherapy, with the number of patients per study ranging from 14 to 119 (details are presented in Table 1).

**Table 1 ijms-26-02620-t001:** Studies included in the general quantitative analysis.

Author; Year	Country	Study Type	RTC Phase	N Interv Group	N Control Group	Line of Therapy	Age (±SD)	Sex M (% M)	PS (ECOG)	Stage	ICI Type	Combined Treatment
Chen 2023 [23]	Asia	RO		27	0	I	64 (46–77)	18 (66.7%)	0–1	IV	Anti-PD-1 (Pembrolizumab; Nivolumab; Toripalimab; Sintilimab; Tislelizumab; Camrelizumab)	GnP
Cheng 2023 [9]	USA, Asia	RO		27	26	I	64 (55–79)	16 (53.3%)	0–1	III, IV	Anti-PD-1 (Sintilimab; Camrelizumab; Tislelizumab; Pembrolizumab)	GnP
Gong 2022 [24]	Asia	RO		104	0	≥I	63 (30–80)	62 (59.6%)	0–2	III, IV	Anti-PD-1 (Pembrolizumab; Camrelizumab; Toripalimab; Sintilimab; Tislelizumab)	CT, targeted therapy, neoantigen vaccine therapy, with or without RT
Lemech 2023 [25]	Australia	RCT	I	18	0	I, II	62 (45–71)	8 (44.4%)	0–1	IV	Anti-PD1 (nivolumab)	Pixatimod
Ma 2019 [26]	Asia	RO		22	36	≥I	56 (34–73)	13 (59.1%)	0–1–2–3	IV	Anti-PD1 (Pembrolizumab; Nivolumab) or anti-PDL1 (Atezolizumab)	CT
Melisi 2021 [27]	Europe, USA, Asia	RCT	Ib	42	0	I, II, III	57 (38–81)	17 (40.5%)	NA	IV	Anti-PDL1 (durvalumab)	(TGFβ) receptor inhibitor; galunisertib
Reiss 2022 [28]	USA	RCT	Ib, II	44	0	≥I	65 (54–73)	29 (66%)	0–1	III, IV	Anti-PD1 (nivolumab 240 mg q3w)	Niraparib
				40	0		63 (58–69)	20 (50%)			anti-PD1 (nivolumab 3 mg/kg)	
Renouf 2022 [29]	USA, Canada	RCT	II	119	61	I	64 (29–81)	67 (56.3%)	0–1	IV	antiCTLA4 (tremelimumab) + antiPDL1 (durvalumab)	GnP
Royal 2010 [30]	USA	RCT	II	27	0	≥I	55 (27–68)	15 (55.5%)	0–1–2	III, IV	AntiCTLA4 (ipilimumab)	No
Sun 2018 [31]	Asia	RO		43	0	≥I	56 (35–85)	25 (58.1%)	0–1–2	IV	AntiPD1 or anti PDL1 or antiCTLA4	CT, targeted therapy, ipilimumab
Tsujikawa 2020 [32]	USA	RCT	II	51	42	II	64 (58–69)	37 (73%)	0–1	IV	Anti-PD1 (nivolumab)	GVAX pancreas vaccine, cyclophosphamide and CRS-207
Wang 2024 [33]	Asia	RCT	II	23	0	≥I	65,3 (53–80)	11 (47.8%)	NA	IV	AntiPD1 (sintilimab, toripalimab)	Oxaliplatin + S1
Xie 2020 [34]	USA	RCT	I	14	0	>I	62 (43–80)	7 (50%)	0–1	IV	AntiPDL1 (durvalumab)	SBRT 8 Gy 1 fr
				19	0		60 (43–85)	10 (53%)			AntiPDL1 (durvalumab) + antiCTLA4 (tremelimumab)	SBRT 8 Gy 1 fr
				16	0		60.5 (44–79)	15 (94%)			AntiPDL1 (durvalumab) + antiCTLA4 (tremelimumab)	SBRT 25 Gy 5 fr
Yang 2024 [35]	Asia	RO		43	301		65 (37–78)	30 (69.8%)	0–1–2–3	IV	Anti-PD1 (nivolumab)	CT add-on
				33	301		62 (46–81)	20 (60.6%)				CT concurrent
				16	301		62 (52–73)	7 (43.75%)				No
Zhang 2022 [36]	Asia	RO		17	31	≥I	≤60: 20 (64.5%); >60: 7 (41.2%)	10 (58.8%)	0–1–2	IV	Anti-PD1 (Toripalimab; Camrelizumab; Tislelizumab; Sintilimab)	GnP
Bockorny 2020 [37]	USA, Europe, Asia	RCT	IIa	29	0	≥I	63.9 (46–86)	18 (48.6%)	0–1	IV	Anti-PD1 (pembrolizumab)	BL8040
				22		II	68 (50–83)	13 (59%)				BL8040 + NALIRI + 5FU + leucovorin
Liu 2022 [38]	Asia	RO		32	34	I	61.5 (49.3–66.0)	20 (62.5%)	0–1	IV	Anti-PD1 (sintilimab)	Nab-paclitaxel plus S1
Liu 2024 [39]	Asia	RO		25	27	I	<60: 16 (66.7%); ≥60: 8 (33.3%)	16 (64%)	0–1–2	IV	Anti-PD1 (camrelizumab and sintilimab)	Gemcitabine/nab-paclitaxel + anlotinib
Parikh 2021 [40]	USA	RCT	II	25	0	≥I	60 (32–75)	18 (72%)	0–1	III, IV	Anti-PD1 (nivolumab) + anti-CTLA4 (ipilimumab)	Radiation
Hong 2022 [41]	USA	RCT	II	17	0	≥I	62.5 (32–80)	61 (53.5%)	0–1	IV	Anti-PD1 (nivolumab)	Mogamulizumab
Callahan 2023 [42]	USA	RCT	II	18	0	≥II	66.5 (35–76)	13 (72%)	0–1	IV	Anti-PD1 (nivolumab)	no
				21	0		63.0 (47–79)	11 (52%)	0–1		Anti-PD1 (nivolumab) + anti-CTLA4 (ipilimumab)	no
				30	0		65.0 (31–78)	18 (60%)	0–1–2		Anti-PD1 (nivolumab) + anti-CTLA4 (ipilimumab)	Cobimetinib

RCT: Randomized controlled trial; SD: Standard Deviation; PS: Performance Statis; ICI: immune checkpoint inhibitor; RO: Retrospective Observational; CT: chemotherapy; GnP: gemcitabine plus nab-paclitaxel; IRE: Irreversible Electroporation.

The aggregated mean age was 60.8 years (range of reported means: 27–86 years); most patients had a performance status (PS) of 0–1, with some studies including patients up to PS 2 or 3. Among the 21 studies, 23.1% included patients in second-line (II) therapy, while 69.2% involved patients in later lines of therapy (≥I, >I, or beyond). Among the studies, 72.4% included anti-PD-1 therapies, 6.9% included anti-PD-L1 therapies, 3.4% used only anti-CTLA4 therapy, and 17.2% involved combinations with anti-CTLA4. In this analysis, 41.4% of combinations involved chemotherapy, 17.2% radiotherapy, and 34.5% experimental agents such as Cobimetinib, Mogamulizumab, BL8040, GVAX pancreas vaccine, niraparib, pixatimod, (TGFβ) receptor inhibitor galunisertib, or anlotinib.

The median OS was 8.65 months, with a range of 1.1 to 18.1 months and a Standard Deviation (SD) of 5.34 months. The median PFS was 2.55 months, with a range of 0.9 to 8.1 months and an SD of 2.31 months. Key outcomes revealed a global mean ORR of 16.2% (range: 0–56.3%; SD: 15.2%), while the mean DCR was 50.3% (range: 0–93.8%, SD: 29.3%). Grade-≥3-treatment-related adverse events (TRAEs) were reported in over 60% of studies, with incidence rates ranging from 7.1% to 62.5%.

### 3.1. Meta-Analysis

The forest plot for OS was divided into two groups: immunotherapy (including immunotherapy combined with locoregional treatments such as radiotherapy) and immunotherapy combined with chemotherapy. The forest plot revealed high heterogeneity in both groups and in the overall model, with I^2^ values of 79.9%, 95.5%, and 96.4%, respectively (Figure 2).

In the overall random-effects model, the median OS was 6.96 months (95% CI 4.38–9.53 months). For the PFS analysis, it was necessary to exclude the study by Wang et al. (2024) [33] due to the lack of an upper limit for the 95% confidence interval (CI) of PFS. High heterogeneity was also observed for PFS across studies (I^2^: 84.9% for immunotherapy, 96.5% for the combination group, and 94.9% for the overall model). The median PFS in the random-effects model was 2.97 months (95% CI 1.95–3.99 months) (Figure 3).

The Baujat test identified Yang et al. [35] and Chen et al. [23] as outlier studies for OS, while for PFS, the study contributing most to the heterogeneity and influencing the overall estimate was Yang et al. [35] (Appendix A).

Regarding publication bias, as shown in Figure 4, the funnel plots for both OS and PFS were significantly asymmetric (*p* = 0.0004 and *p* = 0.0006, respectively). Consequently, the “trim-and-fill” method was applied. Following this adjustment, Egger’s test resulted in *p* = 0.3062 for OS and *p* = 0.0725 for PFS.

A sensitivity analysis was also performed using the “leave-one-out” method, highlighting that the primary sources of heterogeneity were Chen et al. [23] and Yang et al. [35] for OS, and Chen et al. [23], Yang et al. [35], Gong et al. [24], and Liu et al. [39] for PFS (Figure 5).

### 3.2. MSI-Driven Studies

The three studies focusing on MSI/dMMR biomarker-driven populations included a total of 65 patients. The mean age across the studies was 61.7 years (range: 24–85), and 54.2% of patients were male. All patients presented either microsatellite instability (MSI) or mismatch repair deficiency (dMMR). Immune checkpoint inhibitors were used exclusively in this cohort, with 34 patients (52.3%) receiving monotherapy, i.e., anti-PD1 or anti-PD-L1 treatment, and 31 patients (47.7%) receiving combinations with ipilimumab or chemotherapy. In this biomarker-defined population, outcomes were notably improved compared to those of the general studies. The estimated mean PFS was 10.7 months, with a range from 2.1 months to 26.7 months. Regarding OS, one study was not included in the calculation as the data were not available (Wang et al.) [33]. However, the available OS mean was 8.35 months with the values ranging from 4.0 months to 12.7 months. The mean ORR was 36.1% (range: 18.2–48.4%) (Table 2).

### 3.3. Phase I-II/II Multi-Tumor Studies

A total of 11 phase I or I/II studies investigating ICIs in multiple malignancies, including unresectable pancreatic adenocarcinoma, was identified [43,44,45,46,47,48,49,50,51,52]. The publication years ranged from 2020 to 2024, with sample sizes varying from 12 to 86 patients overall. In most trials, participants had received ≥1 line of previous therapy, with one study reporting a median of up to four prior regimens [44]. The aggregated mean age was 63.2 years, with an overall range of 19 to 86 years, and the majority had an ECOG performance status of 0–1. Pancreatic cancer subgroups in the studies represented proportions ranging from 2.3% to 44%, with varying inclusion in dose-expansion cohorts. The characteristics of ICIs, dosing, and, where applicable, combination therapies are described in Table 3. Survival outcomes specific to pancreatic cancer were very limited. In the pancreatic population, Zheng et al. reported an ORR of 11.1% and a DCR of 33.3%, while Voisin et al. documented a median PFS of 1.3 months and a median OS of 4.8 months. Safety profiles in the overall population were relatively consistent with expected immune-related toxicities, with grade ≥3 TRAEs ranging from 10.8% [48] to 30.8% [45].

**Table 2 ijms-26-02620-t002:** Studies focusing on MSI/dMMR biomarker-driven populations.

Author; Year	Country	Study Type	RTC Phase	N Interv Group	N Control Group	Age (±SD)	Sex M (% M)	ICI Type	Combined Treatment	Biomarker Driven
Taieb 2024 [53]	Europe	RO		31	0	62.1 (37–82)	17 (54.8%)	AntiPD1 or antiPDL1	CT, ipilimumab	MSI/ dMMR
Marabelle 2020 [10]	Asia, USA, Europe	RCT	II	22	0	60.0 (20–87)	96 (41.2%)	Anti-PD1 (pembrolizumab)	no	dMMR and MSI-H
André 2023 [54]	Europe, Canada	RCT	II	12	0	63 (24–85)	106 (30.5%)	Anti-PD1 (dostarlimab)	no	dMMR and MSI-H and/or POLE-altered

**Table 3 ijms-26-02620-t003:** Phase I or I/II studies investigating ICIs in multiple malignancies.

Author; Year	Country	Stage	ICI Type	Combined Treatment	Total Population	Pancreas-Specific Subgroup (*n*, %)	Outcomes (Global)
Curigliano 2021 [43]	USA, Europe	IV	Anti-PD1 (spartalizumab)	Sabatolimab	86	2 (2.3%)	Response: 6%; lasting 12–27 months
Hedge 2021 [44]	USA	IV	Anti CTLA4 (ipilimumab)	Evofosfamide	22	7 (31.8%)	ORR: 16.7%; DCR: 83.3%
Kitano 2020 [45]	Asia	IV	Anti-PD1 (cemiplimab)	no	13	1 (14.3%)	ORR: 30.8%; DCR: 46.2%; grade ≥ 3 TEAEs: 30.8%
Muik 2022 [46]	Europe	IV	DuoBody-PD-L1×4-1BB	no	61	6 (9.8%)	DCR: 65.6%; DLT: 9.8%
Naing 2023 [47]	USA	IV	Anti-PDL1 (durvalumab)	IDO1 inhibitor epacadostat (25–300 mg twice daily)	34	15 (44.1%)	ORR: 12.0%; Grade ≥ 3 TRAEs: 20.6%
Papadopoulos 2022 [48]	USA, Australia, Asia	IV	Anti-PD1 (pembrolizumab)	MK-4166 1.1 to 900 mg Q3W	65	8 (12.3%)	ORR: 61.5% (95% CI, 31.6–86.1); DCR: 61.5% (95% CI, 31.6–86.1); Grade ≥ 3 TRAEs: 10.8%
Yamamoto 2024 [50]	Asia	IV	Anti-PDL1 (atezolizumab)	AMY109 14/45 mg/kg Q3W	20	7 (35%)	ORR: 5% (95% CI, 0.1–24.9); DCR: 30% (95% CI, 11.9–54.3); Grade ≥ 3 TRAEs: 15%
Zamarin 2020 [51]	USA	IV	Anti-PDL1 (durvalumab)	Mogamulizumab 1 mg/kg	12	12 (100% dose expansion)	ORR: 5.3% (95% CI, 0.1–26.0); mOS: 8.9 (95% CI, 4.3–18.4); mPFS: 1.9 (95% CI; 1.7–4.4)
	USA		Anti-CTLA4 (tremelimumab)				ORR: 5.3% (95% CI, 0.1–26.0); mOS: 4.4 (95% CI, 2.5–13.4); mPFS: 1.9 (95% CI, 1.4–3.7)
Zheng 2022 [52]	Asia, Australia	IV	Anti-PD1 (penpulimab)	no	60	11 (18.3%)	ORR: 20% (95% CI, 10.8–32.3); DCR: 45% (95% CI, 32.1–58.4); TRAE: 95.5%
Voisin 2024 [49]	Europe	IV	Anti-PD1 (pembrolizumab)	Xevinapant (100, 150, and 200 mg daily for 14 days on/7 days off)	41	14 (dose expansion)	DCR: 13%; ORR: 3%

RCT: Randomized Controlled Trial; ICI: immune checkpoint inhibitor; CI: confidence interval; ORR: Overall Response Rate; DCR: disease control rate; OS: overall survival; PFS: Progression Free Survival; DLT: Dose Limiting Toxicity; TRAE: treatment-related adverse event.

### 3.4. Quality Assessment

The quality of the included studies was assessed using the NIH Quality Assessment Tool and the Newcastle–Ottawa Scale. Detailed results of the quality assessments are available in the Appendix A.

### 3.5. Discussion

The effectiveness of ICIs in the treatment of unresectable pancreatic cancer has been limited, with a median OS across the 21 studies analyzed of 8.65 months, and a notably low median PFS of 2.55 months, both characterized by extremely wide ranges. Less than one quarter of patients responded to treatment (ORR: 16.2%), while the DCR was 50.3%. The benefit of ICIs appeared more evident in combination therapies compared to monotherapy, although direct comparative data between these strategies are not available. It is important to note that these limited outcomes are also influenced by the inclusion of patients with suboptimal performance statuses (ECOG PS often above 0–1) and those treated in late therapy lines.

The data reported in the meta-analysis suggest that while combining ICIs with chemotherapy may have potential benefits, caution is required in claiming true synergy without robust phase III evidence. Some retrospective data have shown improved outcomes, such as PD-1 inhibitors combined with gemcitabine and nab-paclitaxel, which demonstrated a median OS of 12.8 months [39]. It is not surprising that chemo-immunotherapy combinations have failed to produce convincing results in gastrointestinal malignancies [55], particularly in an unselected patient population without specific clinical or biomolecular characteristics. These results emphasize the need for better patient stratification through biomarker-driven approaches to optimize the efficacy of chemo-immunotherapy regimens in gastrointestinal malignancies, especially in pancreatic cancer, where late diagnosis and rapid disease progression require the selection of a treatment strategy that ensures an early and effective response. However, due to the lack of phase III trials and the absence of comparator arms, the benefits of immunotherapy in the general population of pancreatic cancer patients remain unconfirmed and non-generalizable. Combinations with locoregional treatments such as RT have not shown definitive evidence of synergistic potential. Nevertheless, in some cases, a DCR of up to 72% has been observed, as reported in the study by Gong et al. [24]. The integration of experimental agents represents a highly promising avenue for overcoming resistance mechanisms to ICIs. Among these, Cobimetinib and TGFβ inhibitors have demonstrated the potential to enhance responses to ICIs, with an average ORR of 10%. Compared to standard first-line chemotherapy regimens such as FOLFIRINOX [2] and nab-paclitaxel [3], the median OS reported for immunotherapy in our review appeared lower, aligning more closely with outcomes seen in second-line treatments such as those reported in the NAPOLI-1 trial [5], considering that only two studies [15,31] in our meta-analysis reported data on a population treated exclusively in the first-line setting.

In biomarker-defined populations, such as MSI/dMMR, outcomes seemed better, with a mean PFS of 10.7 months and an ORR of 36.1%, although the data are limited to a few studies with very small sample sizes (65 patients in total). Pembrolizumab has been approved for the treatment in MSI-high pancreatic cancer, based on the results of the Keynote-158 (basket) trial [10]. This approval is reflected in clinical guidelines, which recommend pembrolizumab as a second-line or subsequent therapy for patients with metastatic or unresectable MSI-high tumors [56,57]. However, it is important to note that MSI-high pancreatic cancer represents only a small subgroup, comprising approximately 1–2% of all pancreatic cancer cases [11]. Beyond MSI, other potentially predictive biomarkers, such as PD-L1 expression, are under investigation. Though limited in prevalence in pancreatic cancer, PD-L1 remains a valuable marker for response in some cancers. Recent studies indicate that assessing PD-L1 expression, even in circulating tumor cells or on exosomes, could offer complementary data to help predict immunotherapy response [58,59]. Furthermore, Tumor Mutational Burden (TMB) is another biomarker associated with favorable responses to checkpoint inhibitors across several cancers, though its specific role in pancreatic cancer is still uncertain. Recent analyses have found that pancreatic cancer patients with high TMB and PD-L1 expression may derive clinical benefits from immunotherapy, particularly when combined with MSI positivity [58,60]. This combination of biomarkers may help in more accurately selecting patients who could benefit from immunotherapy in pancreatic cancer, though further studies are needed to confirm these observations. The immunosuppressive tumor microenvironment in pancreatic cancer is a key barrier to the efficacy of ICIs. It is characterized by a dense stromal matrix, driven largely by cancer-associated fibroblasts, and limited T-cell infiltration due to factors such as hyaluronan-rich extracellular matrices and immunosuppressive cells, including myeloid-derived suppressor cells and tumor-associated macrophages [61]. This environment impairs T-cell activation and recruitment, reducing the effectiveness of ICIs [61,62]. Therapies targeting the stromal matrix, such as TGF-β pathway inhibitors, have shown some preclinical promise in improving ICI efficacy by enhancing T-cell infiltration and reducing stromal density [26].

Our study found that ICIs are generally well-tolerated, with grade ≥3 TRAEs reported in an average of 22% of cases. The most common adverse events included dermatitis, colitis, and endocrinopathies, as documented in the studies by Gong et al. [24] and Ma et al. [26].

### 3.6. Study Limitations

A random-effects model was used, highlighting the high heterogeneity across the studies. The study by Yang et al. in the immunotherapy add-on arm had the most significant impact on the estimation of both OS and PFS. This can be explained by the fact that the addition of immunotherapy in a population already responding to chemotherapy selects a prognostically favorable group, with improved survival outcomes. As a result, it is expected to deviate significantly from other groups. For Chen et al. [23], the impact, particularly on OS, may be attributed to the fact that the analyzed population was exclusively treated in the first-line setting, with good performance status (ECOG 0–1) and a gold-standard first-line chemotherapy combination. While the OS data exceed those reported in the registration trial of the chemotherapy combination [3], the small sample size (27 patients) and the observational nature of the study introduce significant biases. These limitations reduce the statistical power and generalizability of the results. Additionally, the exclusively Asian population in this study introduces potential ethnic and biological variability that may not be generalizable to other populations. Despite the application of the ’trim-and-fill’ method reducing the asymmetry observed in the funnel plots, it is important to highlight that the evident heterogeneity among the included studies represents a significant limitation. This factor, known to influence the shape of the funnel plot, may reduce the reliability of the method in distinguishing between publication bias and other sources of asymmetry. The inclusion of observational studies with broad selection criteria and the absence of phase III trials represent a significant limitation of this meta-analysis, contributing to the high heterogeneity observed.

The exclusion of studies with fewer than 10 participants aimed to reduce the risk of small-study bias and limit the impact of potentially unreliable estimates on the overall findings. Nevertheless, phase 1 trials, despite their small sample sizes, were included in the qualitative synthesis to capture valuable safety and dosing data. Their exclusion from the quantitative meta-analysis ensured that the pooled estimates were not disproportionately influenced by studies with limited power, maintaining the validity of the results. Among the early-phase studies included, on which it was not possible to perform a quantitative analysis but only a qualitative one, no head-to-head comparisons were available, and most trials enrolled small, heavily pretreated cohorts of patients with advanced pancreatic cancer. These trials, many of which report limited outcome data (e.g., only objective response rates or short-term toxicity), still highlight the potential antitumor activity of ICIs in a setting with few therapeutic options. Moreover, certain endpoints like prolonged disease control in a subset of patients [43] suggest that even in heavily treated populations, immunotherapy could offer incremental benefits. These findings illustrate that, despite the inherent limitations of phase I/II designs—especially single-arm cohorts—such trials can provide valuable insights into safety and exploratory efficacy signals.

## 4. Conclusions

This systematic review systematically analyzed the use of ICIs in the treatment of unresectable pancreatic adenocarcinoma, a disease with an extremely poor prognosis and limited therapeutic options. The findings, derived from a comprehensive synthesis of available literature, confirmed that ICIs do not provide a survival benefit in unselected patient populations, either as monotherapy or in combination with chemotherapy or other experimental agents. Despite significant advancements achieved with ICIs in other malignancies, their efficacy in metastatic pancreatic cancer remains absent outside of biomarker-defined subgroups. In patients with MSI-H or dMMR, ICIs showed more promising results, with a higher median OS and an ORR of 36.1%. These findings have led to recommendations in clinical guidelines for this specific subset of patients. Nevertheless, this population accounts for only 1–2% of pancreatic cancer cases, limiting the broader applicability of these treatments.

Further research is essential to establish the benefits of immunotherapy in the general pancreatic cancer population. Efforts should prioritize the development of novel therapies capable of overcoming the barriers imposed by the immunosuppressive tumor microenvironment and identifying additional biomarkers to improve patient responsiveness.

### Clinical Implications and Future Directions

The routine use of ICIs in clinical practice requires careful patient selection, considering the need for adequate biopsy material to assess biomarkers such as dMMR/MSI. However, the low prevalence of MSI in pancreatic adenocarcinoma patients limits the utility of routine MSI testing. A more effective strategy might involve conducting MSI testing in selected populations with specific clinical or pathological characteristics. From an economic perspective, the cost–benefit ratio of this approach remains a critical factor to consider.

Well-designed, larger RCTs are essential to define the role of ICIs in the treatment of pancreatic cancer. Moreover, the exploration of novel biomarkers is crucial for identifying patients who may derive greater benefit from ICIs and for effectively integrating these biomarkers into clinical practice. Combination strategies, including experimental therapies, continue to hold significant promise for enhancing the efficacy of ICIs and overcoming resistance mechanisms.

Additionally, future research should focus on identifying cost-effective and practical ways to integrate biomarker-driven approaches into routine care. Given the challenges in generalizing results to the broader pancreatic cancer population, studies must also account for real-world patient heterogeneity. While MSI-H patients and those refractory to chemotherapy represent the most evident candidates for ICIs, a further refinement of predictive factors is needed to expand their applicability. In this context, defining optimal treatment sequencing and identifying additional molecular subgroups with immune responsiveness will be crucial for shaping future clinical trial designs. These advances will pave the way for a more personalized approach to immunotherapy, maximizing its potential impact on patient outcomes.

## Figures and Tables

**Figure 1 ijms-26-02620-f001:**
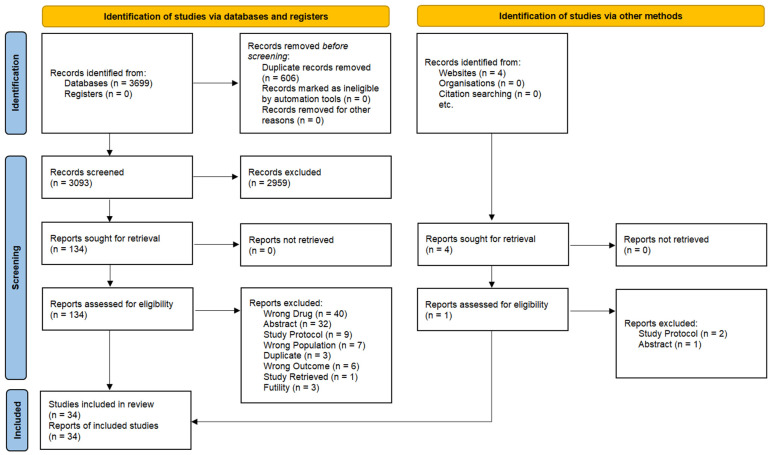
Prisma 2020 flow diagram [14].

**Figure 2 ijms-26-02620-f002:**
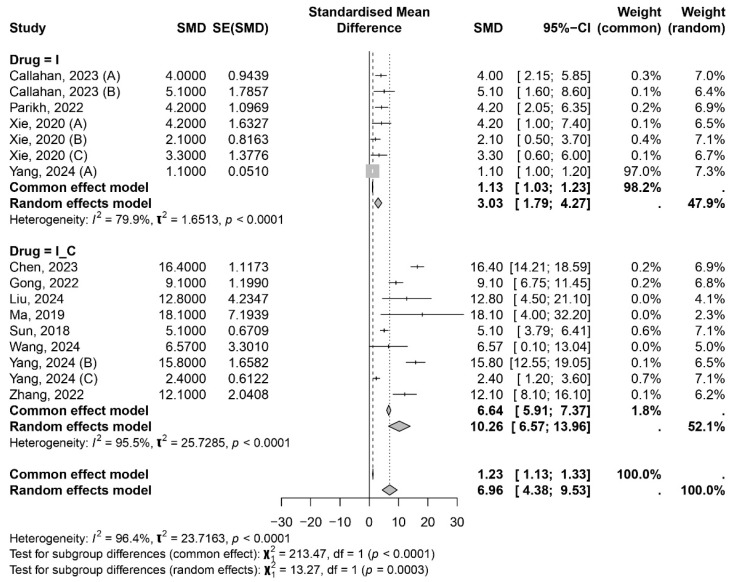
Forest plot displaying the meta-analysis results for OS. Callahan, 2023 (A), (B) [42]; Parikh, 2022 [40]; Xie, 2020 (A), (B), (C), [34]; Yang, 2024 (A) [35]; Chen, 2023 [23]; Gong, 2022 [24]; Liu, 2024 [39]; Ma, 2019 [26]; Sun, 2018 [31]; Wang, 2024 [33]; Yang, 2024 (B), (C) [35]; Zhang, 2022 [36].

**Figure 3 ijms-26-02620-f003:**
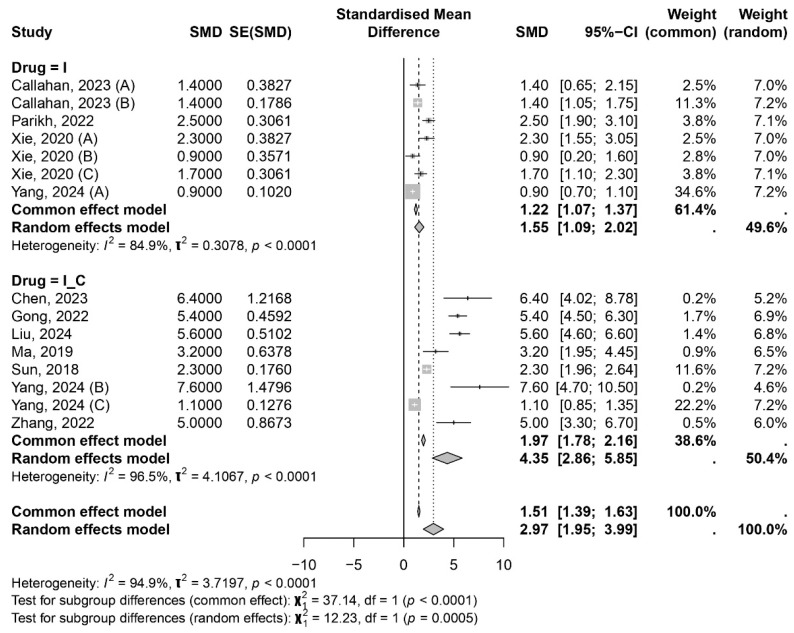
Forest plot displaying the meta-analysis results for PFS. Callahan, 2023 (A), (B) [42]; Parikh, 2022 [40]; Xie, 2020 (A), (B), (C), [34]; Yang, 2024 (A) [35]; Chen, 2023 [23]; Gong, 2022 [24]; Liu, 2024 [39]; Ma, 2019 [26]; Sun, 2018 [31]; Yang, 2024 (A), (B), (C) [35]; Zhang, 2022 [36].

**Figure 4 ijms-26-02620-f004:**
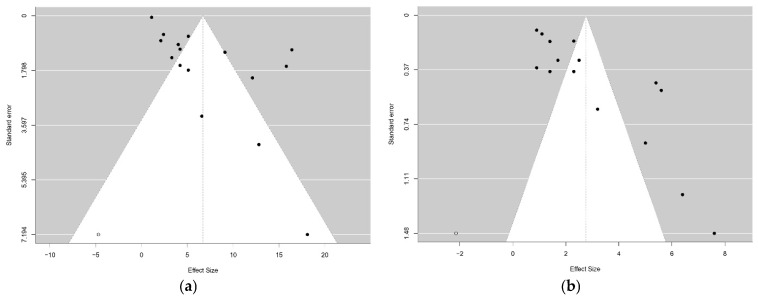
(**a**) Funnel plot illustrating the trim-and-fill method for the OS analysis. (**b**) Funnel plot illustrating the trim-and-fill method for the PFS analysis. The gray dots represent potential missing studies identified by the method, while the black dots correspond to the included studies.

**Figure 5 ijms-26-02620-f005:**
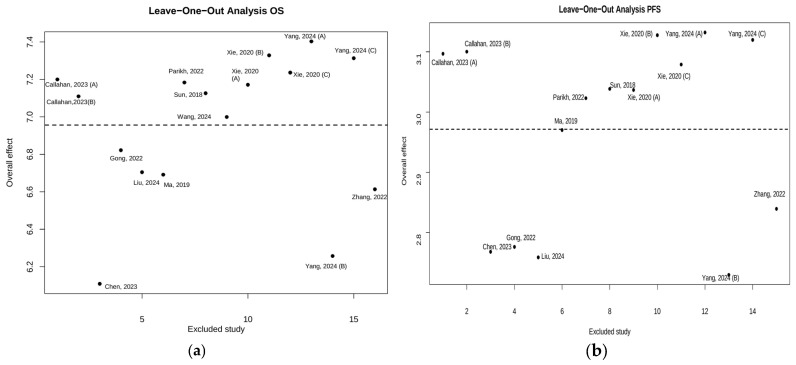
(**a**) Plots of leave-one-out analyses for the OS; Callahan, 2023 (A), (B) [42]; Parikh, 2022 [40]; Xie, 2020 (A), (B), (C), [34]; Yang, 2024 (A) [35]; Chen, 2023 [23]; Gong, 2022 [24]; Liu, 2024 [39]; Ma, 2019 [26]; Sun, 2018 [31]; Yang, 2024 (B), (C) [35]; Zhang, 2022 [36]; Wang, 2024 [33]. (**b**) plots of leave-one-out analyses for the PFS; Callahan, 2023 (A), (B) [42]; Parikh, 2022 [40]; Xie, 2020 (A), (B), (C), [34]; Yang, 2024 (A) [35]; Chen, 2023 [23]; Gong, 2022 [24]; Liu, 2024 [39]; Ma, 2019 [26]; Sun, 2018 [31]; Yang, 2024 (A), (B), (C) [35]; Zhang, 2022 [36].

## Data Availability

The data supporting the finding of this systematic review are available upon request from the corresponding author.

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
