# Peer review of "State of the Art of Immune Checkpoint Inhibitors in Unresectable Pancreatic Cancer: A Comprehensive Systematic Review"

_ijms, 2025, doi:10.3390/ijms26062620_

Round 1
Reviewer 1 Report
Comments and Suggestions for Authors
The review provides a systematic analysis of existing literature, outlining key findings, methodologies, and challenges associated with ICIs in pancreatic cancer treatment. However, it needs a more critical interpretation of the findings. The reported median OS (8.65 months) and PFS (2.55 months) are modest, suggesting that ICIs are not yet a breakthrough in pancreatic cancer. The authors should explicitly compare these results to standard chemotherapy regimens (e.g., FOLFIRINOX or gemcitabine/nab-paclitaxel) to clarify whether ICIs offer any real survival advantage in broader patient populations.
While combination therapies (e.g., ICIs + chemotherapy) show some potential, the review should be more cautious in suggesting synergy without clear phase III evidence. The authors should also discuss why certain ICI-chemotherapy combinations have failed in past clinical trials (e.g., CheckMate-577) and what this means for future research directions.
The review provides a solid scientific foundation but lacks direct clinical applicability. The authors could better define which patients should receive ICIs today—for example, MSI-H patients or chemotherapy-refractory cases—and propose concrete next steps for trial design.
Minor issues:
"4 Department of General Surgery…", "4 Department of Pharmacy…"
Title: "uresectable pancreatic cancer" → "unresectable pancreatic cancer"
Tables are cluttered: Improve spacing between column headers and data rows.
Overall, while systematic reviews and meta-analyses are valuable for summarizing data, this study does not provide new insights or a compelling clinical direction. Instead, it mostly reinforces what is already known—that ICIs alone are ineffective in pancreatic cancer, and chemo-ICI combinations show only marginal improvements with no clear phase III validation.
Author Response
Comments 1: The review provides a systematic analysis of existing literature, outlining key findings, methodologies, and challenges associated with ICIs in pancreatic cancer treatment. However, it needs a more critical interpretation of the findings. The reported median OS (8.65 months) and PFS (2.55 months) are modest, suggesting that ICIs are not yet a breakthrough in pancreatic cancer. The authors should explicitly compare these results to standard chemotherapy regimens (e.g., FOLFIRINOX or gemcitabine/nab-paclitaxel) to clarify whether ICIs offer any real survival advantage in broader patient populations.
Response 1: Dear Reviewer, Thank you for your precise and insightful comments on our work. We acknowledge that the reported median OS is modest; however, this finding is influenced by the treatment line analyzed. In fact, only two of the studies included in our meta-analysis focused exclusively on first-line treatment, making a direct comparison with standard first-line regimens such as FOLFIRINOX or gemcitabine/nab-paclitaxel less appropriate. As this is a systematic review based on studies where immunotherapy was not directly compared with first-line chemotherapy, definitive conclusions on its advantage in this setting cannot be drawn. Currently, there are no phase III studies in the literature directly comparing immunotherapy with standard regimens in pancreatic adenocarcinoma. However, considering a more comparable therapeutic context, such as second-line treatment, the comparison with the NALIRI + 5-FU regimen (NAPOLI-1 trial) appears more relevant. In this study, the median OS was 6.2 months, a value closer to that reported in the studies included in our meta-analysis.
To address this point, we have added the following sentence to the discussion:
"Compared to standard first-line chemotherapy regimens such as FOLFIRINOX (2) and nab-paclitaxel (3), the median OS reported for immunotherapy in our review appears lower, aligning more closely with outcomes seen in second-line treatments such as those reported in the NAPOLI-1 trial (5), considering that only two studies (15,31) in our meta-analysis reported data on a population treated exclusively in the first-line setting."
Comments 2: While combination therapies (e.g., ICIs + chemotherapy) show some potential, the review should be more cautious in suggesting synergy without clear phase III evidence. The authors should also discuss why certain ICI-chemotherapy combinations have failed in past clinical trials (e.g., CheckMate-577) and what this means for future research directions.
Response 2: We appreciate the reviewer’s suggestion. In response, we have revised our discussion on chemo-immunotherapy combinations, adopting a more cautious approach in highlighting their potential benefits. As suggested, we have included a concrete example of a failed chemo-immunotherapy combination, particularly in an unselected patient population, reinforcing the importance of better patient stratification through biomarker-driven approaches.
“The data reported in the meta-analysis suggest that while combining ICIs with chemotherapy may have potential benefits, caution is required in claiming true synergy without robust phase III evidence. Some retrospective data have shown improved outcomes, such as PD-1 inhibitors combined with gemcitabine and nab-paclitaxel, which demonstrated a median OS of 12.8 months [39]. It is not surprising that chemo-immunotherapy combinations have failed to produce convincing results in gastrointestinal malignancies [55], particularly in an unselected patient population without specific clinical and biomolecular characteristics. These results emphasize the need for better patient stratification through biomarker-driven approaches to optimize the efficacy of chemo-immunotherapy regimens in gastrointestinal malignancies, especially in pancreatic cancer, where late diagnosis and rapid disease progression require the selection of a treatment strategy that ensures an early and effective response."
Comments 3: The review provides a solid scientific foundation but lacks direct clinical applicability. The authors could better define which patients should receive ICIs today—for example, MSI-H patients or chemotherapy-refractory cases—and propose concrete next steps for trial design.
Response 3: We have revised the conclusion as requested, incorporating the considerations highlighted in your feedback.
“Well-designed, larger RCTs are essential to define the role of ICIs in the treatment of pancreatic cancer. Moreover, the exploration of novel biomarkers is crucial for identifying patients who may derive greater benefit from ICIs and for effectively integrating these biomarkers into clinical practice. Combination strategies, including experimental therapies, continue to hold significant promise for enhancing the efficacy of ICIs and overcoming resistance mechanisms.
Additionally, future research should focus on identifying cost-effective and practical ways to integrate biomarker-driven approaches into routine care. Given the challenges in generalizing results to the broader pancreatic cancer population, studies must also account for real-world patient heterogeneity. While MSI-H patients and those refractory to chemotherapy represent the most evident candidates for ICIs, further refinement of predictive factors is needed to expand their applicability. In this context, defining optimal treatment sequencing and identifying additional molecular subgroups with immune responsiveness will be crucial for shaping future clinical trial designs. These advances will pave the way for a more personalized approach to immunotherapy, maximizing its potential impact on patient outcomes.”
Comments 4: Department of General Surgery…", "4 Department of Pharmacy…"
Title: "uresectable pancreatic cancer" → "unresectable pancreatic cancer"
Tables are cluttered: Improve spacing between column headers and data rows.
Response 4: We have addressed the minor revisions as suggested.
Reviewer 2 Report
Comments and Suggestions for Authors
This systematic review by Orlandi et al, focused on the analysis of 34 studies addressing the effect of treatments based on immune checkpoint inhibitors (ICI) in the context of pancreatic cancer. The review targets and interesting topic within the treatment landscape for this type of tumor and the analysis has been applied with rigor. The manuscript is well written, and the concepts are clearly presented.
Only a few concerns should be taken into account.
- The authors excluded from the analysis 2959 studies out of 3093 based on predefined eligibility criteria. That is a very important number of studies. Could the authors describe some of these criteria (if not all) in the text. It will be also important to include in Fig 1 legend, the meaning of “*” and “**”.
- In Figure 1, could also authors clarify why the final number of included studies is 34? It seems that from the branch of “Identification of studies via other methods” 1 additional study is included because from the initial 4, only 3 were excluded.
- In section 2.2, it will be helpful to specify the type of monotherapy related to this sentence: “with 34 patients (52.3%) receiving monotherapy and 31 patients (47.7%) receiving combinations with ipilimumab or chemotherapy.
- In the discussion, it will be important to verify if ORR was 16.2% in the case of 21 studies on ICI but also for studies exclusively focused on Cobimetinib TGFBeta inhibitors. This coincidence sounds a little strange.
Author Response
Comments 1: The authors excluded from the analysis 2959 studies out of 3093 based on predefined eligibility criteria. That is a very important number of studies. Could the authors describe some of these criteria (if not all) in the text. It will be also important to include in Fig 1 legend, the meaning of “*” and “**”.
Response 1: Thank you for your insightful comment. We verified that our Number Needed to Read (NNR), as defined in the Cochrane Handbook for Systematic Reviews of Interventions, which refers to the number of studies that must be screened to include one in the final review, was appropriate. An NNR greater than 100 is considered excessively sensitive and reaches the point of 'bibliographic futility.' In our study, 34 studies were included out of 3093 initially screened, based on predefined eligibility criteria (which are detailed in the Materials and Methods section). This results in an NNR of approximately 91, which remains below the threshold of 100.
As suggested, we have clarified this aspect in the Results section. Specifically, the sentence has been modified as follows:
"3093 studies underwent screening, of which 2959 were excluded based on predefined eligibility criteria. In particular, the main reasons for study exclusion during the screening process were background article, wrong outcome, wrong population, wrong publication type and wrong study design."
We acknowledge a typographical error regarding the asterisks in Figure 1. These symbols were originally part of the PRISMA-template flow chart and have now been removed accordingly.
Comments 2: In Figure 1, could also authors clarify why the final number of included studies is 34? It seems that from the branch of “Identification of studies via other methods” 1 additional study is included because from the initial 4, only 3 were excluded.
Response 2: We have identified and corrected a miscalculation in the number of excluded studies. Specifically, the number of studies excluded due to "wrong population" was 7 instead of 6, which now correctly accounts for the final inclusion of 34 studies.
Comments 3: In section 2.2, it will be helpful to specify the type of monotherapy related to this sentence: “with 34 patients (52.3%) receiving monotherapy and 31 patients (47.7%) receiving combinations with ipilimumab or chemotherapy.
Response 3: Thank you for your suggestion. The sentence has been corrected to specify the type of monotherapy, clarifying that it refers to anti-PD1 or anti-PD-L1 treatment. The revised version is: "With 34 patients (52.3%) receiving monotherapy, i.e., anti-PD1 or anti-PD-L1 treatment, and 31 patients (47.7%) receiving combinations with ipilimumab or chemotherapy."
Comments 4: In the discussion, it will be important to verify if ORR was 16.2% in the case of 21 studies on ICI but also for studies exclusively focused on Cobimetinib TGFBeta inhibitors. This coincidence sounds a little strange.
Response 4: We acknowledge that there was a typographical error in the reported ORR. The correct value is 10%, not 16.2%. We have now corrected the sentence as follows:
"Among these, Cobimetinib and TGFβ inhibitors have demonstrated potential to enhance responses to ICIs, with an average ORR of 10%."
Round 2
Reviewer 1 Report
Comments and Suggestions for Authors
This review provides a systematic and data-rich summary of ICIs in pancreatic cancer treatment. However, while it successfully compiles clinical trial data, it lacks a critical interpretation of the findings and does not offer strong clinical takeaways.
The current manuscript title may give a misleading impression that there have been significant advances in using ICIs for unresectable pancreatic cancer, which is not supported by the data in the review.
Author Response
Comments 1: This review provides a systematic and data-rich summary of ICIs in pancreatic cancer treatment. However, while it successfully compiles clinical trial data, it lacks a critical interpretation of the findings and does not offer strong clinical takeaways.
Response 1: Dear reviewer, as this is a systematic review, our conclusions are strictly based on the available literature, analyzed systematically with a reproducible methodology. In light of the current evidence, immunotherapy in metastatic pancreatic adenocarcinoma has not demonstrated a survival benefit in unselected patient populations, whether used as monotherapy or in combination with chemotherapy or other experimental agents. While ICIs have transformed the therapeutic landscape in several malignancies, the data reviewed in our study confirm that their efficacy in pancreatic cancer remains limited to highly selected biomarker-driven subgroups, such as those with microsatellite instability or mismatch repair deficiency.
In response to this feedback, we have refined the manuscript’s conclusion:
“This systematic review systematically analyzed the use of ICIs in the treatment of unresectable pancreatic adenocarcinoma, a disease with an extremely poor prognosis and limited therapeutic options. The findings, derived from a comprehensive synthesis of available literature, confirm that ICIs do not provide a survival benefit in unselected patient populations, either as monotherapy or in combination with chemotherapy or other experimental agents. Despite significant advancements achieved with ICIs in other malignancies, their efficacy in metastatic pancreatic cancer remains absent outside of biomarker-defined subgroups. In patients with MSI-H or dMMR, ICIs showed more promising results, with a higher median OS and an ORR of 36.1%. These findings have led to recommendations in clinical guidelines for this specific subset of patients. Nevertheless, this population accounts for only 1–2% of pancreatic cancer cases, limiting the broader applicability of these treatments.”
Comments 2: The current manuscript title may give a misleading impression that there have been significant advances in using ICIs for unresectable pancreatic cancer, which is not supported by the data in the review.
Response 2: We acknowledge that the previous title, "Advances in Immune Checkpoint Inhibitors for Unresectable Pancreatic Cancer: A Comprehensive Systematic Review", may have unintentionally suggested that significant progress has been made in the field, whereas our findings indicate that the clinical impact of immune checkpoint inhibitors remains limited, except in specific biomarker-driven subgroups. To more accurately reflect the scope and conclusions of our review, we have modified the title to:
"State of the art of immune checkpoint inhibitors for unresectable pancreatic cancer: a Comprehensive Systematic Review"
This revision ensures that the title remains comprehensive while avoiding any misleading implications of substantial breakthroughs. We believe this modification aligns more precisely with the systematic review’s findings and methodology, and we hope the reviewer finds this acceptable.